# Living with Psychosis without Mental Health Services: A Narrative Interview Study

Rose McGranahan,[1] Zivile Jakaite ![ORCID],[1] Alice Edwards,[2] Stefan Rennick-Egglestone,[3] Mike Slade,[3] Stefan Priebe[1]

[1]Unit for Social and Community Psychiatry (WHO Collaborating Centre for Mental Health Service Development), Queen Mary University of London, London, UK
[2]Newham Centre for Mental Health, East London NHS Foundation Trust, London, UK
[3]School of Health Sciences, Institute of Mental Health, University of Nottingham, Nottingham, UK

**Correspondence to**
Zivile Jakaite;
zivile.jakaite@nhs.net

## ABSTRACT

**Objectives** Little research has looked at how people who do not use mental health services experience psychosis. Thus, the present study aimed to explore the experiences and views of people with psychosis who have neither sought nor received support from mental health services for at least 5 years.

**Design** A narrative interview study. Thematic analysis was used to analyse the data.

**Setting** England.

**Participants** Twenty-eight participants with self-defined psychotic experiences were asked to provide a free narrative about their experiences.

**Results** Five themes were identified: (1) Perceiving psychosis as positive; (2) Making sense of psychotic experiences as a more active psychological process to find explanations and meaning; (3) Finding sources of strength, mainly in relationships and the environment, but outside of services; (4) Negative past experiences of mental health services, leading to disengagement and (5) Positive past experiences with individual clinicians, as an appreciation of individuals despite negative views of services as a whole.

**Conclusions** Perceiving psychosis as something positive, a process of making sense of psychotic experiences and the ability to find external sources of strength all underpin—in addition to negative experiences with services—a choice to live with psychosis outside of services. Future research may explore to what extent these perceptions, psychological processes and abilities can be facilitated and strengthened, in order to support those people with psychosis who do not seek treatment and possibly also some of those who are in treatment.

## INTRODUCTION

Research has shown that a considerable number of people with psychosis do not present to mental health services.[1] According to one review, 7% of the general population are likely to have psychotic experiences in their lifetime.[2] Other reviews estimate voice-hearing prevalence at 13%[3] and experience of paranoia in up to 30% of the general population.[4] Since these rates are much higher than the number of patients with psychosis in health services,[5] many people with such

### Strengths and limitations of this study

► To our knowledge, this study is one of the first to explore the experiences of people with psychosis outside of services who are hard to reach and commonly not included in research studies.
► A free narrative approach emphasised the perspective of the interviewees and allowed for rich material to be analysed.
► The participants included in this study were recruited from a wide-range of organisations and geographical areas within England, thus enabling a maximum variation within the target population.
► The sample was heterogeneous in terms of biographies and living situations, but it was a convenience sample and exclusively from England.
► The study did not entail formal diagnostic assessments.

experiences are likely to live with and manage their experiences without conventional treatment or other support from mental health services.

There is a debate as to whether the psychosis-like experiences in the general population are qualitatively distinct from those in clinical populations.[6 7] Peters *et al* [8] found that both clinical and non-clinical groups with psychotic experiences experienced positive symptoms. Interestingly, the non-clinical group had lower levels of paranoia, delusions, cognitive difficulties and negative symptoms, but more somatic or tactile hallucinations.

Previous literature exploring the views of people with psychotic-like experiences who do not seek support from mental health services is scarce. To our knowledge, one prior study conducted by Boumans *et al* [9] documented two key factors in maintaining well-being without having to access mental health services: personalised self-care and adoption of interpretive frameworks (in order to fully make sense of psychotic experiences).

Studying the experiences of people with psychosis outside of services can be

challenging, as many people in such situations may avoid not only contact with health services, but also participation in research studies. However, such research may: identify different strategies for living with psychosis than those found in clinical samples; point towards forms of support that are appropriate for people with psychosis who do not seek treatment; and provide evidence on attitudes, skills and resources that can promote recovery in other groups.[10] Against this background, the current study aimed to explore the experiences and views of people with psychotic experiences who have not received any treatment or other support from mental health services for the past 5 years.

## METHODS
### Study design
We conducted qualitative semi-structured narrative interviews. The narrative methodological approach was chosen as it enables a flexible collection of rich material while simultaneously placing emphasis on the perspective of the interviewees.[11] A Lived Experience Advisory Panel advised on the content and conduct of the interviews, including the phrasing of questions and the terminology used in the interviews.

As per the ethics application, the study had a predefined sample size of up to 30 participants. It was agreed that participant recruitment would be terminated after data saturation was reached.

### Participants and recruitment activities
Participants met the following inclusion criteria: having had self-defined experiences of what could be termed psychosis (with examples given of hearing voices or seeing or believing things that others do not), not having accessed mental health services for the previous 5 years although they may have had accessed them earlier (ie, secondary or tertiary mental health services in the categorisation of the National Health Service in the UK, that is, services providing treatment through fully qualified mental health professionals), over 17 years of age, and able to give informed consent.

Recruitment activities were structured and iteratively refined to seek a maximum variation sample of psychosis experiences.[12] To achieve a maximum variation sample of psychosis experiences, participants from a wide range of organisations and geographical areas (in and outside of London) were recruited. The recruitment of participants took place through: primary care (general practitioner practices) services in London, online support groups or networks (eg, spiritual or critical psychiatry networks), groups affiliated to the Hearing Voices Network, and charity and non-governmental organisations. It is important to highlight that the organisations included in the current study do not provide specialist mental health treatment or support. Although the recruitment method varied slightly depending on the site being recruited from, in most instances the recruitment was facilitated

by presentations and attendance at a variety of groups, advertising online or sending invitation letters to potential participants.

Researchers explained the details of the study on the phone to potential participants, and encouraged them to ask questions or raise concerns. This was followed by a meeting in person, in which individuals could make their final decision as to whether to take part in the research interview. All participants received a participant information sheet and were given a sufficient amount of time to consider whether to participate in the study. As part of the recruitment process, all of the eligibility criteria were discussed with each participant to verify that they were being met. There was no prior relationship between the researchers conducting the interviews and the study participants.

### Procedure
Twenty-eight semistructured interviews were conducted by three researchers (RM, ZJ and JLB) over a 15-month period, with most interviews (n=25) carried out by RM. Most interviews were conducted in London, with some in other parts of England. Each participant took part in a 40–120 min interview conducted in health services or community venues. On the day of the interview, researchers provided the potential participants with an in-depth explanation of the study procedures and aims, ensuring their full understanding and ability to provide informed consent. All participants provided written informed consent. During the interviews, the researchers asked open-ended questions designed to elicit a narrative,[13] with minimal or no interruption from the researcher in order to facilitate fluent sharing of experiences.[14] Specifically, participants were asked to share their recovery story, followed by some questions focusing on participants' experiences of sharing their recovery story with others. All participants were reimbursed for their participation.

### Analysis
Interviews were audiorecorded, transcribed verbatim, anonymised and were then transferred to NVivo V.11 for data management and analysis. After the researchers familiarised themselves with the content of the data, an initial inductive thematic analysis was conducted to identify key themes across the interviews, based on an approach outlined by Braun and Clarke.[15] Preliminary codes were established from this analysis in relation to the research question, and refined through a continuous discussion in the wider research team, from backgrounds in clinical and academic psychiatry and clinical and academic psychology. Transcripts were then analysed in more detail by two members of the research team, with 20% being double coded and checked for consistency by two independent researchers. Discrepancies were discussed until resolved. Coding consisted of identification and allocation of text relating to the preliminary coding framework, allowing comparison of themes occurring within

and across sources. Once all transcripts were coded, the lead author synthesised the coding into themes with the two analysts. Regular discussions were held between all analysts to adapt and develop the themes, which underwent several iterations from its original form to the final version. The researchers maintained reflexivity by consistently having discussions within the team and with the wider research group to gain a more varied perspective. Reflexive field notes were also utilised which focused on the researcher's role as an interviewer, including their feelings, possible biases and immediate context prior to the interview in order to further ensure reflexivity.

### Patient and Public Involvement
A Lived Experience Advisory Panel advised on the content and conduct of the interviews.

## RESULTS
### Sample
Twenty-eight people with self-defined psychotic experiences were interviewed between February 2018 and May 2019. The duration of the interviews ranged from 40 to 120 min. Eleven participants were recruited via primary care practices and 17 through alternative networks and advertising. Sixteen were female, and 11 male (1 preferred not to say). Four participants reported being in the age group of 25–34 years; six in the group of 35–44 years; nine in the group of 45–54 years; four in the group of 55–64 years and two being more than 64 years of age (one preferred not to say). Eighteen self-declared ethnicity as White, two as Asian and five as Black (three preferred not to say).

### Themes
Five distinct themes were identified.

### Perceiving psychosis as positive
A large proportion of participants perceived their experience of psychosis as positive overall.

> I think I have something extraordinary. And people with extraordinary things find it hard in society, and probably I don't find it hard' (A01)

For some, this related to seeing such experiences, for example, their hallucinations, as a positive coping mechanism to deal with difficult life events, even if the hallucinations themselves could be difficult:

> …my brain was trying to process all the things happening around me, and tried to externalise it in some form that I could perhaps deal with a bit better… (A27)

Others found their experiences educational or empowering, and were able to learn or gain new and positive perspectives from them:

> I kind of tried to get rid of all the negativity and then ended up getting like a higher kind of version of

myself so I felt really empowered and really kind of connected to the universe (A02)

> I feel that in that space I get some real truths (A010)

For some, their psychosis had become a positive part of their identity, an essential part of how they saw themselves in the world, providing a resource that they could contribute:

> I have taken the only course open to me to cope with symptoms, that is to treat seeing things which are not there, or hearing what nobody else does, as attributes which in truth are a part of who I am and harnessing them as the strengths they really are, and the gifts which I have to give the world. (A19)

> …in a way I see it as a positive, they don't have to be negative, you can actually accept who you are and… you make the best of it really (A020)

### Making sense of psychotic experiences
When people managed to make sense of their psychotic experiences, this often also implied a positive view of the experiences. It was, however, not just a positive perspective of what happened. It often involved a more active psychological process of developing a meaningful explanation. Indeed, the process of making sense of the psychotic experiences was also often linked to the biographical context of the individual.

An explanation for why they had experienced psychosis was important, and these explanations themselves became a significant part of their narratives.

> I have since learnt that actually there is meaning to things and actually I am somebody that…does seem to have a perception or radar that not everybody seems to have…in one way the psychosis was awful, but in another way it exposed this horrible past that was unknown, unexplored, unexpressed (A07)

Many linked their experiences with negative life events, such as abuse in childhood or difficult upbringing:

> …I've been a voice hearer and someone who hallucinates since I was little…the reason for that in my particular case is because I am a survivor of incest. (A05)

> …I come from a fairly dysfunctional family…It did actually take me years and years to put together the idea… that the mental health problems sort of arose as…a means for my brain to externalise what was going on inside… sometimes they (hallucinations) were more supportive than the people around me were… maybe my brain created an enviroment for me…to be in…I understood better…I had control of (A027)

These negative experiences could be from a period before individuals were able to form memories, so that finding out about their past could make sense of their experiences:

> And that's when I found out that we'd been separated when I was a very, very young baby…so you know and

my soothing- self-soothing system probably wasn't that well developed, so that helped me to understand (A11)

Other negative experiences were from later in life:

…my mental health challenges were triggered by having radical surgery. I mean it is - it is important to say this, so, before (the surgery) I hadn't experienced (mental health challenges)… (A16)

Some attributed their experiences to spiritual reasons, finding them empowering and meaningful:

This story is in my ancestry, that's my theory of what happened, I don't have any other explanation and I felt that there was- I felt that's what happened to me (A10)

These explanations were sometimes linked to religious beliefs:

…as a Christian I don't see anything that ever happens to anybody as being a result of random chance anymore, I think it was important to my journey (A22)

### Finding sources of strength

A third theme was the sources of strength that participants found around them, in relationships and in their natural environment, but outside of mental health services.

These included both receiving and giving social support, and the validation and feelings of connection provided by others, particularly those with similar experiences:

…it was amazing, we all kind of met up and we had all been through similar things, and it was incredible to just hear everyone speak and be open (A02)

The significance of social support was also expressed in terms of the importance of close personal relationships:

Well I think my partners have been too important in my life. So the first time when I got my diagnosis I was with my boyfriend… at that time he was very, very, very supportive and so was his family. (A09)

Some people spoke about the importance of taking ownership of their narrative for their recovery, creating a coherent sequence of events, often through counselling or alternative therapy work:

…part of the reason I find narrative therapy actually so interesting as an approach is that part of the recovery process has been assembling the story. And understanding it as a story. (A06)

Others found the process of creating a written narrative cathartic:

…I wrote my autobiography so that was a big part of my healing process, it was- It was a catharsis I needed to look back and understand my journey… that… was a huge part of my healing process. (A11)

For some, opposing the dominant illness model of psychosis and using spiritual or other alternative frameworks to make sense of their experiences was essential to being well:

…going into the different reality… and taking it seriously rather than dismissing it as illness was my recovery, that was what helped me to recover, I think. (A10)

Many people spoke about the importance of self-care in managing their mental health, and finding ways to improve their well-being on a daily basis. For example, people found exercise and healthy eating to be helpful:

I started to exercise, and I'd started to clean up my food, and I started to do a bit of meditation… And I said, oh, wow, this is how you get better then. (A16)

Other forms of self-care mentioned were making time and space for oneself, and spending time in nature:

I love the sea, every time I feel like anxious and nervous I just look at the sea and it is just, like, beautiful. (A02)

Another helpful factor people spoke about was finding purpose in daily life, and productive ways to use their time. These could entail creative outlets such as music and art forms:

…it's recovery through my artwork every day… —I think the art has given me great kind of- great kind of structure… (A01)

For others, purpose was achieved by engaging in or returning to higher education or seeking to educate themselves:

…it was important to me sort of coming back to schooling, that was… unfinished business because I could never go… (A04)

Others found purpose in voluntary or paid employment:

…the job that I do is what keeps me out of hospital, I know it yeah, and it has done for all that time because I haven't been hospitalised… (A12)

### Negative past experiences of mental health services

Some participants had previously engaged with services, and several of them reported negative experiences of treatments and interactions with staff, which contributed to their decision to disengage from health services. These could relate to unwanted medication side effects:

The medication didn't help at all, actually it numbed me so much that I had no point any more (A04)

But also to coercive and forceful treatment by staff:

I remember they were dragging me up the ward… to the seclusion room, where they monitored me…they were very, very forceful and all I said was I just need

your help and……it wasn't really dealt with, no-one answered back, that was the process. (A021)

The regime was punitive…if you…stepped out of line… the notion was you would receive more intensive care which of course was nonsense, it was a form of punishment. (A020)

And indeed abuse, as reported by two participants:

Unfortunately I was raped in hospital and it made it—made it very uncomfortable to be there so I left really quickly. (A03)

…when I was sectioned um in the local mental health hospital I was raped, probably by staff, which added to my trauma… (A05)

### Positive past experiences with individual clinicians

However, past experiences with services were mixed. Those people who reported positive experiences often referred to the importance of helpful relationships with individual clinicians, rather than a team or a whole service, so that these positive interactions did not provide a motivation to re-engage with the service. The individual clinicians made them feel heard, and were able to see them as a whole person as opposed to the sum of their symptoms:

When I said to him (psychiatrist), he said 'tell me about yourself' I said 'there it is, read this' he pushed my file away and said 'that's a piece of paper- it tells me nothing about you' and then we connected, we clicked (A17)

Others spoke about clinicians who did not discount the validity of their experiences and their potential meaning, respecting their interpretations of experiences:

…I had a couple of sessions with a psychiatrist… … and I actually said to him 'do you think it's possible to see, like have an episode like a psychotic episode and in that episode literally see the things that you're going to experience in your life all in that episode and then everything actually begin very, very slowly' and he said 'it's possible, you know'. (A08)

Even participants with negative views of services as a whole, tended to distinguish between individual clinicians. The participant who criticised a punitive regime in the previous theme also remarked:

I thought the psychiatrist was very pleasant, we had some interesting chats…(after leaving)…I felt the loss of the attention of the psychiatrists… (A020)

## DISCUSSION

We explored the experiences and views of people living with psychotic experiences, who have neither sought nor received any support from mental health services in the last 5 years. The focus was on understanding how people describe their views and experiences when not receiving treatment or any other type of help from mental health professionals. Five themes were identified in the narratives. Three themes reflect views and abilities that may be essential for the participants' choice for not contacting services. They include a positive perception of psychotic experiences even if they are seen as unusual, an active and—at least in parts—successful search for sense and meaning, and a link with the external world in form of finding strength in personal and natural sources around them. Two further themes address the past experiences with services, which were often negative but also involved helpful relationships with individual clinicians.

### Strengths and limitations

While numerous studies have investigated how patients with diagnosed psychotic disorders experience and are satisfied or dissatisfied with the treatment they have been receiving in mental health services, this is one of the first studies to explore the narratives of a hard-to-reach population of people with experiences of psychosis who are not in contact with mental health services. The methodological approach of encouraging people to tell their story in a free narrative emphasised the perspective of the interviewees and allowed for rich material to be analysed. We recruited from different groups, and reached saturation for generating the main themes.

The study also has several limitations. Firstly, we recruited a convenience sample. Given that there may be very different groups of people with the experience of psychosis who do not use mental health services, other samples may have yielded a different picture. Secondly, although the sample was selective, it was still heterogeneous in terms of biographies and living situations. However, it was overall too small to analyse differences between subgroups. Thirdly, the approach of a free narrative limited the option to focus on specific aspects in more depth and detail. And finally, neither did the interviewers conduct a formal diagnostic assessment nor did they check the diagnostic assessment of possible treatment episodes in the past. Thus, one may question to what extent experiences of the interviewees would be classified as psychosis if such formal assessments had been made.

### Summary of the results and comparison against the literature

A previous study investigating the experiences of people with psychotic symptoms not accessing mental health services[9] found themes relating to self-care and the importance of underlying explanatory frameworks. Similarly, two themes in this study centre on the importance of individuals' positive perceptions of their experiences and finding a meaningful explanation. Perceiving psychotic experiences as meaningful and related to their life situation has been shown to be beneficial for people's prospects of recovery.[16]

The findings suggest that it is important that people's explanations are aligned with and accepted by others, as well as contain personal meaning. An aspect of helpful

social support was sharing a joint explanatory model for their experiences, and receiving validation of these beliefs. In addition, we found that individual relationships with clinicians could be particularly helpful when individuals felt that they were listened to, and their explanations were taken seriously. This is in line with previous research showing that sharing the same explanatory model for psychosis is associated with improved treatment satisfaction and therapeutic relationship in outpatients diagnosed with schizophrenia[17] and more generally with a large body of evidence showing the importance of the direct communication between patients and individual clinicians.[18] The importance of establishing a positive patient-clinician relationship through appropriate communication is underlined by the fact that many participants still expressed their appreciation for individual clinicians, although these relationships had ceased at least 5 years ago and all participants had decided not to seek help from mental health services in general anymore.

Many of the participants believed that specific traumatic experiences had played a causal role in the development of psychotic experiences. Previous narrative reviews and meta-analyses have suggested a link between traumatic experiences and psychosis,[19–21] which may be influenced by lower psychosocial functioning and greater social adversity.[8]

Having social connections was considered helpful not only in relation to the validation of explanatory models, but also for daily support and well-being, which is consistent with research showing the important role of social networks for recovery.[22 23] In addition, people spoke about the importance of being engaged in occupations, religious groups and creative activities. Such activities may provide purpose and structure.[24] Having structure and using various forms of self-care to stay well have been found to be associated with larger social networks.[25]

That people who prefer not to be in contact with services report a range of negative past experiences with services is not surprising. Indeed, the fact that negative experiences can lead to disengagement has been shown before[26] and may be seen as obvious. Some people reported damaging and traumatic events such as being raped which can easily be understood as a reason to avoid any contact with services from then onwards. Overall, however, the negative reports about experiences with services were not more pronounced than in some studies with people who are still—or again—in treatment.[26 27] Explanatory models for their experiences that are not shared, the feeling of not being listened to or treated with respect, and further stressful events during treatment, particularly on wards, are also frequently found in accounts of patients who are in the care of mental health services.[28 29] Moreover, people in this study repeatedly described very positive experiences with individual clinicians although this did not generalise to positive views of the services as a whole. Thus, one may speculate as to whether the negative experiences reported in this study are rather common and possibly less important than the positive views and

resources in relation to the decision not to seek treatment as well as the ability to manage psychotic experiences outside of services. Seeing psychotic experiences—for various reasons—as something positive, making sense of the experiences and finding external sources of strength might be more specific for this group of people than the critical views of health services.

## Conclusions

Given a lack of previous literature, the current study sought to explore the experiences and views of people with psychotic experiences who have neither sought nor received support from mental health services for at least 5 years by employing a free narrative interview methodology.

However, it remains an open question as to whether any and, if so, how many of the interviewees might actually have benefitted from whatever support mental health services in the given area can provide. Services that accommodate and respect a range of explanatory models and have clinicians with good communication skills who treat patients with interest and respect should be more likely to maintain helpful relationships even with people who have different explanations for their experiences. This might even outweigh some negative treatment experiences that many patients might have at some stage of their long-term pathways.

A challenge for future research is to explore whether it is possible to facilitate and strengthen the positive perceptions, psychological processes and abilities to find strength in engaging with external sources, that were identified in this study. Such research might lead to better support for people with psychosis who do not seek treatment. It might also benefit the further development of approaches for people with psychosis who are in treatment, as it could help their recovery, although in some patients it might also undermine the motivation to stay in treatment.

**Acknowledgements** This paper is independent research funded by the National Institute for Health Research (NIHR) under its Programme Grants for Applied Research Programme (Programme Grants for Applied Research, Personal Experience as a Recovery Resource in Psychosis: NEON Programme, RP-PG-0615-20016). MS acknowledges the support of the Center for Mental Health and Substance Abuse, University of South-Eastern Norway, and the NIHR Nottingham Biomedical Research Centre. The views expressed are those of the authors and not necessarily those of the NIHR or the Department of Health and Social Care. The authors would like to thank Joy Llewellyn-Beardsley for assistance with interview data collection.

**Contributors** SP designed the study and led the analysis, MS was in charge of the overall research programme and SR-E co-ordinated data collection across sites. The interviews were conducted by RM and ZJ whilst all the transcripts were read by RM, ZJ, AE and SP. The analysis of the data and discussions regarding the codes/themes were undertaken by RM, ZJ, AE and SP. RM wrote the first draft of the manuscript. All of the coauthors have contributed towards revisions of the manuscript and have approved the final version of it.

**Funding** The research was funded by the NIHR under its Programme Grants for Applied Research Programme (RP-PG-0615–20016).

**Disclaimer** The views expressed are those of the authors and not necessarily those of the NIHR.

**Competing interests** None declared.

**Patient and public involvement** Patients and/or the public were involved in the design, or conduct, or reporting, or dissemination plans of this research. Refer to the Methods section for further details.

**Patient consent for publication** Not required.

**Ethics approval** This research was undertaken as part of the NIHR Narrative Experiences Online (NEON) study (ISRCTN11152837, information at http://www.researchintorecovery.com/neon) between March 2018 and August 2019. Ethical Committee approval was obtained in advance (Nottingham 2 REC 17/EM/0401). All participants provided written informed consent.

**Provenance and peer review** Not commissioned; externally peer reviewed.

**Data availability statement** No additional data are available.

**ORCID iD**
Zivile Jakaite http://orcid.org/0000-0002-0013-6827

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
