## [Reviewer comments · BMJ Open]

ARTICLE DETAILS

TITLE (PROVISIONAL)	Living with Psychosis without Mental Health Services: A Narrative Interview Study
AUTHORS	McGranahan, Rose; Jakaite, Zivile; Edwards, Alice; Rennick-Egglestone, Stefan; Slade, Mike; Priebe, Stefan

VERSION 1 – REVIEW

REVIEWER	Temesgen, Worku A Bahir Dar University College of Medical and Health Sciences
REVIEW RETURNED	31-Oct-2020

GENERAL COMMENTS	Participants inclusion criteria and the definition are a bit conflicting; it was already determined participants were those “who have not received any treatment or other support from mental health services for the past five years” But participants were recruited from “primary care ... services in London, online support groups or networks (e.g. spiritual or critical psychiatry networks), presentations and attendance at groups affiliated to the Hearing Voices Network, and advertising via charity and non-governmental organisations.” Despite I am not familiar with the health care system of the UK, these networks are specific for those people with mental illness (psychosis) which mean they were/are getting mental health services one way or the other, they might not get direct antipsychotic medications but ... it seems these networks might be assisted by professionals (mental health or spiritual experts). Perhaps, it would be better if a brief description about these networks/groups is given so that readers can make their own judgment. It feels there are some discrepancies between how authors defined participants as “naïve to psychiatric treatment or not in mental health care” and the actual data in the manuscript where it seems participants are those who either completed or disengaged from hospital treatment and switched to a community group. It feels if the themes are too disperse and loose, maybe tightening-up them to a couple of themes possibly with sub-themes can make the overall finding more robust. Otherwise, the whole narratives in the result are rich in content demonstrating the beauty of qualitative study The word “Psychosis” is a medical term so needs proper medical/psychiatric diagnosis to label a person as “Living with Psychosis”; better if authors could find other less stringent term. “Lived Experience Advisory Panel ad ...” their role in this study is unclear. There should be something describing interview questions (guides). How to assure if participants were not receiving any mental health service for the past 5 years, and how did you actually know if they
---

	had/were having psychosis experience? (did you used any screening question/s? Anyone refused to participate? I do not think reflexivity can be assured with “continually returning to the data to maintain reflexivity” “Eleven were recruited via primary care practices and 17 through alternative networks and advertising.” This perhaps needs more elaboration, for what purpose did participants visited “primary care practices”; and reasons of being member of those networks. About samples, it would be better if you give more information about the clinical characteristics, (duration with symptom, if history of treatment, any other illness (diagnosed) ... so that the reflexivity in this regard.
--	--

REVIEWER	Arnold, Chelsea Australian Catholic University
REVIEW RETURNED	23-Nov-2020

GENERAL COMMENTS	This paper seeks to understand the experiences of a difficult to reach population – people with experiences of psychosis who have not sought or received mental health treatment. Understanding recovery narratives and broader experiences of this group is an important and under researched area. This study is novel and includes some interesting themes. The results have the potential to inform improvements to mental health service delivery and outcomes for people with psychosis. Despite this merit, it is my opinion that this article has several substantial issues and requires significant revision to warrant acceptance for publication. I outline my specific concerns below:  - Please refer to the Journal Article Reporting Standards for Qualitative Research (JARS-Qual; Levitt et al., 2018) and incorporate the requirements throughout the manuscript. I have outlined several areas for improvement below. However, the authors should refer to these standards and incorporate the requirements throughout. - The research question is not consistent throughout the manuscript. Please clarify who the participant group of interest is and ensure the research question is consistent. Introduction:  - Please ensure that references throughout the manuscript are appropriate. For example, references [1] and [5] appear to be inappropriately used. Kreyenbuhl et al. (2009) reviews disengagement from mental health services, which is distinct from not presenting to services. Kendler et al. (1996) examines factors associated with non-affective psychosis in the community, and does not appear to provide information on the number of people with psychosis in health services. - Please include an overview & critique of relevant literature on this topic to justify the current study. There is some reference to previous literature in the discussion but this is lacking in the introduction. As such, the rationale for the current study is lacking. - Please also provide a rationale for the use of qualitative methods/the narrative approach taken. Methods: There is a significant amount of detail missing from the methods. In accordance with reporting standards for qualitative research (Levitt et al., 2018) please include the following information:  - Please make it clear how participant inclusion criteria were assessed. For example, ‘self-defined experiences of what could be termed psychosis’ – was this assessed in any way? How did the
--

	research team determine whether participants were able to give informed consent?  - Please provide further information regarding recruitment approach – how were participants identified and approached? - Please provide further information regarding sampling. How did the authors 'seek a maximum variation sample of psychosis experiences'? - Please describe the process for determining the number of participants and the rationale for the decision to halt data collection. - Please also note whether there was any relationship between participants and the researcher. - Please note whether the participants were reimbursed for their participation. - Please outline in more detail the narrative approach taken – how does this fit with the research question? - Please provide some example of questions asked to 'elicit a narrative' - How did the Lived experience advisory panel advise on the content and conduct of the review? - Please provide further detail of the analytic approach. Braun & Clarke is provided as a reference but the stages taken do not appear to align with the steps of thematic analysis. - Please provide a description of how reflexivity was used during data collection & analysis. Results:  - As per reporting guidelines, please provide the average length of interviews. - Did the authors collect any additional information about the sample? It would be could to include any additional information. For example, it would be particularly helpful to know whether the participants had previous exposure or engagement with mental health services (outside the 5 year) - The themes included appear largely descriptive. Further analysis to determine the underlying shared meaning (in accordance with thematic analysis; Braun & Clarke (2006, 2020) would strengthen the results. For example, the first theme 'Perceiving psychosis as positive' is quite broad and doesn't convey a strong sense of the meaning underpinning the theme. The second theme 'making sense of psychotic experiences' appears to have some overlap with the first theme – thus, further review of themes may be required. - The aim of the study is to examine recovery narratives – however this doesn't appear prominent in the results. The concept of recovery narratives could be better introduced in the introduction to provide some context for this. For example it isn't clear how the theme 'negative past experience of mental health services' is a recovery narrative. Discussion:  - There is a well-balanced discussion of results. However, the conclusion does not appear to link well to the introduction or aims of the study.
--	---

VERSION 1 – AUTHOR RESPONSE

Comment	Response	Location
---------	----------	----------

2. REVIEWER 1:

2.1 Participants inclusion criteria and the definition are a bit conflicting; it was already determined participants were those “who have not received any treatment or other support from mental health services for the past five years”

But participants were recruited from “primary care ... services in London, online support groups or networks (e.g. spiritual or critical psychiatry networks), presentations and attendance at groups affiliated to the Hearing Voices Network, and advertising via charity and non-governmental organisations.” Despite I am not familiar with the health care system of the UK, these networks are specific for those people with mental illness (psychosis) which mean they were/are getting mental health services one way or the other, they might not get direct antipsychotic medications but ... it seems these networks might be assisted by professionals (mental health or spiritual experts). Perhaps, it would be better if a brief description about these networks/groups is given so that readers can make their own judgment. It feels there are some discrepancies between how authors defined participants as “naïve to psychiatric treatment or not in mental health care” and the actual data in the manuscript where it seems participants are those who either completed or disengaged from hospital treatment and switched to a community group.

We appreciate that the system may be unclear for people from outside of the UK and thanks for pointing it out. We have clarified this in the text.

Just to clarify, primary care services and alternative networks are not part of the specialist mental health services within the National Health Service in the UK. Both do not provide specialist mental health treatment.

The study sought to recruit participants who have ‘not accessed mental health services for the previous five years although they may have *had accessed them earlier* (i.e. secondary or tertiary mental health services in the categorisation of the National Health Service in the United Kingdom).’

For instance, primary care services include general practitioners which are typically seen as a ‘front door’ to the health service and approximately 85% of the UK population reports seeing a general practitioner within the last 12 months, whilst alternative networks are typically groups which although vary hugely in terms of their activities and structure, also do not provide specialist mental health treatment.

Lines 144-145.

	To make it more clear, we added a sentence to the manuscript explaining that both primary care services and alternative groups do not provide specialist mental health treatment.	
2.2 It feels if the themes are too disperse and loose, maybe tightening-up them to a couple of themes possibly with sub-themes can make the overall finding more robust.	We have gone through a long iterative process and felt that the themes are sufficiently distinct and represent five distinct areas. However, as per the suggestion made by Reviewer 2, we made the distinction between themes 1 and 2 clearer.	Lines 258-259.
2.3 The word “Psychosis” is a medical term so needs proper medical/psychiatric diagnosis to label a person as “Living with Psychosis”; better if authors could find other less stringent term.	Although the use of the term ‘Psychosis’ is contentious, the term is widely used and accepted in the literature. We felt that the use of the word ‘Psychosis’ is appropriate in the context of this study as it was used during the process of participant recruitment.	No action required.
2.4 “Lived Experience Advisory Panel ad ...” their role in this study is unclear.	The Lived Experience Advisory Panel was asked to feedback on the wording of the questions in the interview, and which terms should be used. Two members of the panel also took part in a researcher training day, where the researchers practiced their interviewing techniques with the panel members, and they fed back on ways to make the experience as comfortable as possible. As per your suggestion, we have added some additional information regarding the role of the Lived Experience Advisory Panel to the manuscript.	Lines 119-121.

2.5 There should be something describing interview questions (guides).	A description of the interview questions has been added to the methods section.	Lines 171-173.
2.6 How to assure if participants were not receiving any mental health service for the past 5 years, and how did you actually know if they had/were having psychosis experience? (did you used any screening question/s? Anyone refused to participate?	As part of the recruitment process, the eligibility of the participants was checked prior to the participation by discussing each eligibility criteria with the potential participants and ensuring participants met it, including the use of any mental health services within the last 5 years and experiences of psychosis-related experiences. Although the recruitment method slightly varied depending on the site being recruited from, in most cases the study was advertised either online or by sending letters. Potential participants were asked to contact the research team to further discuss the research study and to express their interest to participate. So for this reason, we have not approached anyone who has not volunteered to take part in the study. Also, all participants received a participant information sheet and were given a sufficient amount of time to consider whether to participate in the study. We have added some more information regarding this to the methods section.	Lines 156-158.
2.7 I do not think reflexivity can be assured with “continually	We apologise for the wording of this, certainly it could have been	Lines 192-196.

returning to the data to maintain reflexivity”	done better. We have reworded this sentence to reflect how the research team maintained reflexivity. For instance: The research team met frequently for reflexive discussions and reflexive field notes were recorded and utilised by the researchers. The notes focused on: (1) Reflection on researcher’s role as an interviewer including feelings, possible biases as well as immediate context for oneself prior to the interview. (2) Reflection on own role as a participant in the narrative: researcher’s responses to participant, changes in questions, any additional questions asked.	
2.8 “Eleven were recruited via primary care practices and 17 through alternative networks and advertising.” This perhaps needs more elaboration, for what purpose did participants visited “primary care practices”; and reasons of being member of those networks.	The study did not collect information regarding the reasons behind participants’ use of primary care services /groups/networks. It was not possible to explore this as the ethics of the study would not permit to. However, as also mentioned previously, majority of the population is registered with a general practitioner in the UK and 85% of the population report seeing a general practitioner within the last year.	No action required
2.9 About samples, it would be better if you give more information about the clinical characteristics, (duration with symptom, if history of treatment, any other illness	We did collect some demographic information which was reported in the manuscript. However, we are not able to report any further information.	No action required

(diagnosed) ... so that the reflexivity in this regard.	One of the main strengths of this study is the fact that it sought to recruit a-hard-to-reach-population. However, this does not come without it's challenges. Since participants were recruited from a range of non-medical organisations, we did not have access to medical notes (nor would our ethics have permitted us to access medical notes of the participants). Moreover, the clinical characteristics in the form of standardised assessments have not been undertaken. We felt it would have been inappropriate in the context of this study. Majority of the people taking part are very critical regarding the concept of diagnosis, symptoms and assessments. Many would have refused to complete such assessments or to participate altogether. This in turn would have resulted in a selection bias and therefore we decided not to undertake such assessments.	
3. REVIEWER 2:		
3.1 Please refer to the Journal Article Reporting Standards for Qualitative Research (JARS-Qual; Levitt et al., 2018) and incorporate the requirements throughout the manuscript. I have outlined several areas for improvement below. However, the authors should refer to these standards and	Thanks for suggesting this paper. We have reviewed the guidelines and incorporated the standards throughout the manuscript as per the comment.	N/A

incorporate the requirements throughout.		
3.2 The research question is not consistent throughout the manuscript. Please clarify who the participant group of interest is and ensure the research question is consistent.	Apologies for the inconsistent phrasing. As per your suggestion, the research question/participant group have been reviewed and clarified, ensuring that it is consistent throughout the manuscript. The study aimed to explore the experiences and views of people with psychotic experiences who have not received any treatment or other support from mental health services for the past five years.	Lines: 40-41; 393;409-411.
3.3 Please ensure that references throughout the manuscript are appropriate. For example, references [1] and [5] appear to be inappropriately used. Kreyenbuhl et al. (2009) reviews disengagement from mental health services, which is distinct from not presenting to services. Kendler et al. (1996) examines factors associated with non-affective psychosis in the community, and does not appear to provide information on the number of people with psychosis in health services.	We have reviewed the references. Regarding the Kreyenbuhl et al. (2009) [1] review, although it is not explicitly clear from the title, the paper actually also reviews the literature to date about 'individuals who fail to seek mental health treatment at all'. For this reason, we felt this is an appropriate reference. In terms of the Kendler (1996), we have replaced it with a more appropriate reference - (Public Health England, 2016).	Reference section
3.4 Please include an overview & critique of relevant literature on this topic to justify the current study. There is some reference to previous literature in the discussion but this is lacking in the introduction. As such, the rationale for the current study is lacking.	We aimed to keep the introduction concise. Research is scarce regarding this topic. As per your comment, we have noted the relevant prior research in the introduction section in order to highlight the rationale for the current study.	Lines 98-103.

3.5 Please also provide a rationale for the use of qualitative methods/the narrative approach taken.	A brief sentence has been added to the introduction to justify the use of qualitative methods in the context of this study.	Lines 117-119.
3.6 There is a significant amount of detail missing from the methods. In accordance with reporting standards for qualitative research (Levitt et al., 2018) please include the following information:  - Please make it clear how participant inclusion criteria were assessed. For example, 'self-defined experiences of what could be termed psychosis' – was this assessed in any way? How did the research team determine whether participants were able to give informed consent? 	Regarding the eligibility screening, as part of the recruitment process, the eligibility of the participants was checked prior to the participation by discussing each eligibility criteria with the potential participants and ensuring participants met it. In terms of the ability to provide informed consent, on the day of the interview, researchers provided the potential participants with an in-depth explanation of the study procedures and aims, ensuring their full understanding and ability to provide informed consent. We have added additional information to the methods section to clarify this.	Lines: 166-168; 156-178.
3.7 Please provide further information regarding recruitment approach – how were participants identified and approached?	We have added some additional information regarding the recruitment strategy. Although the recruitment method varied slightly depending on the site being recruited from, in most instances the recruitment was facilitated by presentations and attendance at a variety of groups, advertising online or sending invitation letters to potential participants.	Lines 146-160.

	Researchers explained the details of the study on the phone with potential participants, and encouraged them to ask any questions or raise concerns. This was followed by a meeting in person, in which individuals could make their final decision as to whether to take part in the research interview.	
3.8 Please provide further information regarding sampling. How did the authors 'seek a maximum variation sample of psychosis experiences'?	We aimed to achieve a maximum variation within the sample by recruiting participants from a wide-range of organisations (primary care, non-governmental organisation, networks and groups) and geographical areas (in and outside of London) which we felt would provide us with a representative and comprehensive summary of the views and experiences within the target population. Some additional information regarding participant recruitment and sampling has been added to the manuscript.	Lines 138-140.
3.9 Please describe the process for determining the number of participants and the rationale for the decision to halt data collection.	In our field of work, when applying for an ethics approval we have to pre-specify the number of participants that the study will aim to recruit. As per the ethics application, the study had a pre-defined sample size of up to 30 participants. The data collection was halted due to data saturation.	Lines 122-124.

	A brief summary of this has been added to the methods section.	
3.10 Please also note whether there was any relationship between participants and the researcher.	There was no relationship between both of the researchers conducting the interviews and the study participants. This has been added in the methods section.	Lines 158-160.
3.11 Please note whether the participants were reimbursed for their participation.	The participants were reimbursed for their participation. This information has been added to the methods section.	Line 173.
3.12 Please outline in more detail the narrative approach taken – how does this fit with the research question?	The narrative methodological approach was chosen as it enables for a flexible collection of rich material whilst simultaneously placing emphasis on the perspective of the interviewees (Greenhalgh, et al., 2005). As per your suggestion, further information has been added regarding the choice of the free narrative approach.	Lines 117-119.
3.13 Please provide some example of questions asked to 'elicit a narrative'	We have added a description of the interview questions to the methods section as per your suggestion. For instance, in the first part of the interview, participants were asked to share their recovery story, followed by questions focusing on their experiences of sharing their recovery story with others.	Lines 171-173.
3.14 How did the Lived experience advisory panel	The Lived Experience Advisory Panel was asked to feedback on the wording of the questions	Lines 119-121.

advise on the content and conduct of the review?	in the interview, and which terms should be used. Two members of the panel also took part in a researcher training day, where the researchers practiced their interviewing techniques with the panel members, and they fed back on ways to make the experience as comfortable as possible. Some additional information regarding the role of the Lived Experience Advisory Panel has been added.	
3.15 Please provide further detail of the analytic approach. Braun & Clarke is provided as a reference but the stages taken do not appear to align with the steps of thematic analysis.	Additional information has been added to the analysis section in order to make it more clear.	Lines 176-196.
3.16 Please provide a description of how reflexivity was used during data collection & analysis.	The researchers maintained reflexivity by consistently having discussions within the research team and the wider research group to gain a more varied perspective. Reflexive field notes were also utilised which focused on researcher's role as an interviewer, including their feelings, possible biases and immediate context prior to the interview in order to further ensure reflexivity. As per your suggestion, we have added additional information to the manuscript.	Lines 192-196.
3.17 As per reporting guidelines, please provide the average length of interviews.	We are not able to report the mean duration of the interviews as this information has not been recorded. However, the researchers undertaking the interviews	Lines 210-211.

	reported that the interviews were never shorter than 40 minutes and did not exceed 120 minutes. We have added this information to the results section.	
3.18 Did the authors collect any additional information about the sample? It would be could to include any additional information. For example, it would be particularly helpful to know whether the participants had previous exposure or engagement with mental health services (outside the 5 year)	Reviewer 1 has also raised a similar point. Additional information was not collected. As also suggested in the previous response: One of the main strengths of this study is the fact that it sought to recruit a-hard-to-reach-population. However, this does not come without it's challenges. Since participants were recruited from a range of non-medical organisations, we did not have access to medical notes (nor would our ethics have permitted us to access medical notes of the participants). Moreover, the clinical characteristics in the form of standardised assessments have not been undertaken. We felt it would have been inappropriate in the context of this study. Majority of the people taking part are very critical regarding the concept of diagnosis, symptoms and assessments. Many would have refused to complete such assessments or to participate altogether. This in turn would have resulted in a selection bias	N/A

	and therefore we decided not to undertake such assessments.	
3.19 The themes included appear largely descriptive. Further analysis to determine the underlying shared meaning (in accordance with thematic analysis; Braun & Clarke (2006, 2020) would strengthen the results. For example, the first theme 'Perceiving psychosis as positive' is quite broad and doesn't convey a strong sense of the meaning underpinning the theme. The second theme 'making sense of psychotic experiences' appears to have some overlap with the first theme – thus, further review of themes may be required.	As per your suggestion, we have further reviewed the themes and made the distinction between theme 1 and theme 2 more clear.	Lines 258-259.
3.20 The aim of the study is to examine recovery narratives – however this doesn't appear prominent in the results. The concept of recovery narratives could be better introduced in the introduction to provide some context for this. For example it isn't clear how the theme 'negative past experience of mental health services' is a recovery narrative.	Thanks for pointing out that the aim of the study was inconsistently worded. We reviewed it throughout the manuscript, ensuring that it's consistent. 'The present study aimed to explore the experiences of people living with psychosis who have neither sought nor received support from mental health services for at least five years using the recovery narrative methodology'.	N/A
3.21 There is a well-balanced discussion of results. However, the conclusion does not appear to link well to the introduction or aims of the study.	The conclusion has been reviewed to ensure it links better with the introduction and the aims of the study.	Lines 489-492.

VERSION 2 – REVIEW

REVIEWER	Temesgen, Worku A Bahir Dar University College of Medical and Health Sciences
REVIEW RETURNED	22-Mar-2021

GENERAL COMMENTS	The manuscript improved much; particularly it cleared unclarities about study participants. Themes are now better phrased, however the main concern I still have is themes strength and their independence from one another. Despite authors insist to have “Five distinct themes” I doubt these themes are truly distinct/independent. I think these themes can be tighten in to two or three. 1st and 2nd themes “1) Perceiving psychosis as positive 2) Making sense of psychotic experiences as a more active psychological process to find explanations and meaning” can be merged into one, 4th and 5th can be also made as one theme. I wish these themes are re-named/labeled in more attractive, with catchy and less words. Authors claimed “first to explore the experiences of people with psychosis outside of services ...” Yes, it may be the first at the study area but there are several reports from different parts of the world. Inclusion criteria ... “over 18,” please edit this. Another main concern persists yet: the report repeatedly emphasized “participants are those who do not have mental health care for at least five years” AND participants were recruited from organizations (what so ever) supporting individuals with some kind of mental health problems. Is it not conflicting? I already raised this concern at the previous version and some corrections are made; despite do not address the issue. Authors argued this saying “It is important to highlight that the organisations included in the current study do not provide specialist mental health treatment or support”. May be this is a contextual issue that I do not understand; does it mean in UK context health services provided by these organization do not considered as mental health service/treatment? What if you change study participants as those who do not take antipsychotics for 5 or more years or something similar rather than referring them “without Mental Health Services”? We know that “mental health service” is too broad and very contextual. Just a suggestion! These were the main issues I raised to the previous version which sustains to the current version. I saw authors reply to these comments and Yes some are convincing. I let the decision to the editor/s. With these restrains, I think the manuscript is well developed and ready for publication.
--

VERSION 2 – AUTHOR RESPONSE

Reviewer: 1

Comment 1: The manuscript improved much; particularly it cleared unclarities about study participants. Themes are now better phrased, however the main concern I still have is themes strength and their independence from one another. Despite authors insist to have “Five distinct themes” I doubt these themes are truly distinct/independent. I think these themes can be tighten in to two or three. 1st and 2nd themes “1) Perceiving psychosis as positive 2) Making sense of psychotic experiences as a more active psychological process to find explanations and meaning” can be

merged into one, 4th and 5th can be also made as one theme.
I wish these themes are re-named/labeled in more attractive, with catchy and less words.

Response: We feel that merging the themes would reduce the value and richness of the findings. For example (and referring to the suggestion of the reviewer), there is a difference between perceiving an experience as positive in the first place, and making sense of an experience in an active psychological process over time. Also, negative and positive experiences of services are not just two sides of the same coin, but distinct phenomena, as we were trying to illustrate in the paper. Of course, there is an overlap of some of the themes. That themes are distinct is not claiming that they are totally 'independent' and in fact we have not used that term in the paper. However, distinguishing between the different positive experiences is precisely what this study adds.

We accept that other authors might have been able to come up with more attractive, catchier and briefer labels for the themes, but the current titles appear correct and clear to us.

Comment 2: Authors claimed "first to explore the experiences of people with psychosis outside of services ...". Yes, it may be the first at the study area but there are several reports from different parts of the world.

Response: We have added, 'to our knowledge' to the mentioned phrase in the summary of strengths and limitations. This leaves the possibility that there are studies that we have not found (mere case reports would be a different matter). However, if the reviewer is aware of such publications in other parts of the world, we would have been grateful for a reference that we could have shared with the readership.

Comment 3: Inclusion criteria ... "over 18," please edit this.

Response: As suggested, this has been corrected and now reads 'over 17 years of age'.

Comment 4: Another main concern persists yet: the report repeatedly emphasized "participants are those who do not have mental health care for at least five years" AND participants were recruited from organizations (what so ever) supporting individuals with some kind of mental health problems.

Is it not conflicting?

I already raised this concern at the previous version and some corrections are made; despite do not address the issue. Authors argued this saying "It is important to highlight that the organisations included in the current study do not provide specialist mental health treatment or support". May be this is a contextual issue that I do not understand; does it mean in UK context health services provided by these organization do not considered as mental health service/treatment? What if you change study participants as those who do not take antipsychotics for 5 or more years or something similar rather than referring them "without Mental Health Services"? We know that "mental health service" is too broad and very contextual. Just a suggestion!

Response: We see that the context of the NHS can be difficult to understand for an outsider. There is in fact no contradiction. In order to explain this even further we added that the secondary and tertiary services in the NHS provide treatment through fully qualified mental health professionals. Beyond that, the reviewer is right in his/her understanding that the voluntary organisations we recruited from do not provide specialised treatments. We hope that the explanation in the text is clearer, and that readers will understand it now. We cannot change the criterion of not using mental health services, because is that is exactly the criterion we used.

We hope that the revised paper is acceptable for publication in BMJ Open.